# Machine Learning Force Fields with Data Cost Aware Training

## Abstract

Machine learning force fields (MLFF) have been proposed to accelerate molecular
dynamics (MD) simulation, which finds widespread applications in chemistry and
biomedical research. Even for the most data-efficient MLFFs, reaching chemical
accuracy can require hundreds of frames of force and energy labels generated by
expensive quantum mechanical algorithms, which may scale as $O(n^3)$ to $O(n^7)$,
with $n$ proportional to the number of basis functions. To address this issue, we
propose a multi-stage computational framework – ASTEROID, which lowers the
data cost of MLFFs by leveraging a combination of cheap inaccurate data and
expensive accurate data. The motivation behind ASTEROID is that inaccurate
data, though incurring large bias, can help capture the sophisticated structures
of the underlying force field. Therefore, we first train a MLFF model on a large
amount of inaccurate training data, employing a bias-aware loss function to prevent
the model from overfitting tahe potential bias of this data. We then fine-tune the
obtained model using a small amount of accurate training data, which preserves the
knowledge learned from the inaccurate training data while significantly improving
the model's accuracy. Moreover, we propose a variant of ASTEROID based on
score matching for the setting where the inaccurate training data are unlabeled.
Extensive experiments on MD datasets validate the efficacy of ASTEROID.

## 1 Introduction

Molecular dynamics (MD) simulation is a key technology driving scientific discovery in fields such
as chemistry, biophysics, and materials science [Alder and Wainwright, 1960, McCammon et al.,
1977]. By simulating the dynamics of molecules, important macro statistics such as the folding
probability of a protein [Tuckerman, 2010] or the density of new materials [Varshney et al., 2008]
can be estimated. These macro statistics are an essential part of many important applications such as
structure-driven drug design [Hospital et al., 2015] and battery development [Leung and Budzien,
2010]. Most MD simulation techniques share a common iterative structure: MD simulations calculate
the forces on each atom in the molecule, and use these forces to move the molecule forward to the
next state.

The fundamental challenge of MD simulation is how to efficiently calculate the forces at each
iteration. An exact calculation requires solving the Schrödinger equation, which is not feasible
for many-body systems [Berezin and Shubin, 2012]. Instead approximation methods such as the
Lennard-Jones potential [Johnson et al., 1993], Density Functional Theory (DFT, Kohn [2019]),
or Coupled Cluster Single-Double-Triple (CCSD(T), Scuseria et al. [1988]) are used. CCSD(T)
is seen as the gold-standard for force calculation, but is computationally expensive. In particular,

Submitted to NeurIPS 2021 AI for Science Workshop.

35  CCSD(T) has complexity $\mathcal{O}(n^7)$ with respect to the number of basis functions used along with a
36  huge storage requirement [Chen et al., 2020]. To accelerate MD simulations while maintaining high
37  accuracy, machine learning based force fields (MLFFs) have been proposed. MLFFs take a molecular
38  configuration as input and then predict the forces on each atom in the molecule, consequently speeding
39  up the force calculation step.

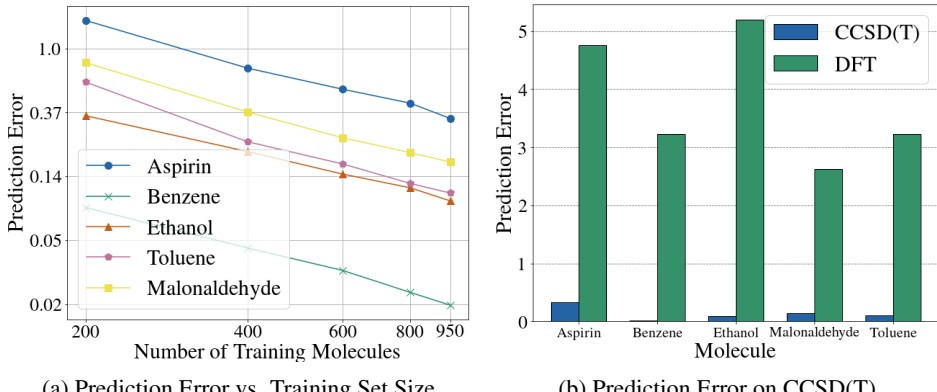

(a) Prediction Error vs. Training Set Size          (b) Prediction Error on CCSD(T)

Figure 1: (a) Log-log plot of the number of training points versus the prediction error for deep force
fields (b) Prediction error on CCSD labeled molecules for force fields trained on large amounts of
DFT reference forces (100,000 configurations) and moderate amounts of CCSD reference forces
(1000 configurations). In both cases the model architecture used is GemNet [Gasteiger et al., 2021].

40  Most recently, deep learning techniques for force fields have been developed, resulting in highly
41  accurate force fields parameterized by large neural networks [Gasteiger et al., 2021, Batzner et al.,
42  2022]. Despite their empirical success, these methods suffer from a critical drawback: *in order to*
43  *train state-of-the-art machine learning force field models, a large amount of costly training data must*
44  *be generated.* For example, to train a model at the CCSD(T) level of accuracy, at least a thousand
45  CCSD(T) calculations must be done to construct the training set. This is computationally expensive
46  due to the method's $\mathcal{O}(n^7)$ cost.

47  A natural solution to this problem is to train on fewer data points. However, if the number of training
48  points is decreased, the accuracy of the learned force fields quickly deteriorates. In our experiments,
49  we empirically find that the prediction error and the number of training points roughly follow a
50  power law relationship, with prediction error $\propto$ (Number of Training Points)$^{-1}$ [Müller et al., 1996,
51  Cortes et al., 1993]. This can be seen in Figure 1a, where prediction error and train set size are
52  observed to have a linear relationship with a slope of $-1$ when plotted on a log scale.

53  Another option is to train the force field model on less accurate but computationally cheap reference
54  forces calculated using DFT [Kohn, 2019] or empirical force field methods [Johnson et al., 1993].
55  However, these algorithms introduce undesirable bias into the force labels, meaning that the trained
56  models will have poor performance. This phenomenon can be seen in Figure 1b, where models
57  trained on large quantities of DFT reference forces are shown to perform poorly relative to force
58  fields trained on moderate quantities of CCSD(T) reference forces. Therefore current methodologies
59  are not sufficient for training force field models in low resource settings, as training on either small
60  amounts of accurate data (i.e. from CCSD(T)) or large amounts of inaccurate data (i.e. from DFT or
61  empirical force fields) will result in inaccurate force fields.

62  To address this issue, we propose to use both large amounts of inaccurate force field data and small
63  amounts of accurate data to reduce the data generation cost needed to achieve highly accurate force
64  fields. Our motivation is that computationally cheap data, though incurring large bias, can help
65  capture the sophisticated structures of the underlying force field. Moreover, if treated properly, we
66  can further reduce the bias of the obtained model by taking advantage of the accurate data.

67  Specifically, we propose a multi-stage computational framework – dat**A** cos**ST** awar**E** t**R**aining of
68  f**O**rce f**I**el**D**s (ASTEROID). In the first stage, small amounts of accurate data are used to identify the

bias of force labels in a large but inaccurate dataset. In the second stage, the model is trained on the large inaccurate dataset with a bias-aware loss function. This loss function generates smaller weights for data points with larger bias, suppressing the effect of label noise on training. The inaccurately trained model serves as a warm start for the third stage, where it is fine-tuned on the small and accurate dataset. Together, these stages allow the model to learn from many molecular configurations while incorporating highly accurate force data, significantly outperforming conventional methods trained with similar data generation budgets.

Beyond using cheap labeled data to boost model performance, we also develop a method for the case where a large amount of unlabeled molecular configurations are cheaply available [Smith et al., 2017, Köhler et al., 2022]. Without labels, we cannot adopt the supervised learning approach. Instead, we draw a connection to score matching, which learns the gradient of the log density function with respect to each data point (called the score) [Hyvärinen, 2005]. In the context of molecular dynamics, we notice that if the log density function is proportional to the energy of each molecule, then the score function with respect to a molecule's position is equal to the force on the molecule. Based on this insight, we show that the supervised force matching problem can be tackled in an unsupervised manner. This unsupervised approach can then be incorporated into the ASTEROID framework, improving performance when limited data is available.

We demonstrate the effectiveness of our framework with extensive experiments on different force field data sets and downstream simulation tasks. We use two popular model architectures, GemNet [Gasteiger et al., 2021] and EGNN [Satorras et al., 2021], and verify the performance of our method in a variety of settings. These experiments show that ASTEROID can lead to significant gains when either DFT reference forces or empirical force field forces are viewed as inaccurate data and CCSD(T) configurations are used as accurate data. In addition, we show that we can learn accurate forces via the connection to score matching, and that using this objective in the second stage of training can improve performance on both DFT and CCSD(T) datasets.

## 2  Background

◇ **Machine Learning Force Fields.** Recent years have seen a surge of interest in MLFFs. Much of this work has focused on developing machine learning architectures that have physically correct equivariances, resulting in large graph neural networks that can generate highly accurate force and energy predictions [Gasteiger et al., 2021, Satorras et al., 2021, Batzner et al., 2022]. Two popular architectures are EGNN and GemNet. Both models are translation invariant, rotationally equivariant, and permutation equivariant. EGNN is a smaller model and is often used when limited resources are available. The GemNet architecture is significantly larger and more refined than the EGNN architecture, modeling various types of inter-atom interactions. GemNet is therefore more powerful and can achieve state-of-the-art performance, but requires more resources to train.

It has been observed that modern MLFFs often cannot achieve sufficient test accuracy to be reliable for MD simulations [Stocker et al., 2022]. Critically, the accuracy of deep force fields such as GemNet and EGNN is highly dependent on the size and quality of the training dataset. With limited training data, MLFFs cannot achieve the required accuracy for usefulness, preventing their application in settings where data is expensive to generate (e.g. large molecules). The amount of resources needed to train is therefore a key bottleneck preventing the widespread use of MLFFs.

◇ **Data Generation Cost.** The training data for MLFFs can be generated by a variety of force calculation methods. These methods exhibit an accuracy cost tradeoff: accurate reference forces from methods such as CCSD(T) require high computational costs to generate reference forces, while inaccurate reference forces from methods such as DFT and empirical force fields can be generated fairly quickly. Concretely, CCSD(T) is highly accurate but has $\mathcal{O}(n^7)$ complexity, DFT is less accurate with complexity $\mathcal{O}(n^3)$, and empirical force fields are inaccurate but could have complexity as low as $\mathcal{O}(n)$ [Lin et al., 2019, Ratcliff et al., 2017]. CCSD(T) is typically viewed as the gold standard for calculating reference forces, but its computational costs often make it impractical for MD simulation (it has been estimated that "a nanosecond-long MD trajectory for a single ethanol molecule executed with the CCSD(T) method would take roughly a million CPU years on modern hardware")

[Chmiela et al., 2018]. Due to this large expense, MLFF training data is typically generated first with MD simulations driven by DFT or empirical force fields. These simulations generate a large number of molecular configurations, and then CCSD(T) reference forces are computed for a small portion of these configurations. Therefore, a large amount of inaccurately labeled molecular configurations are often available along with the accurate CCSD(T) labeled data.

# 3 ASTEROID

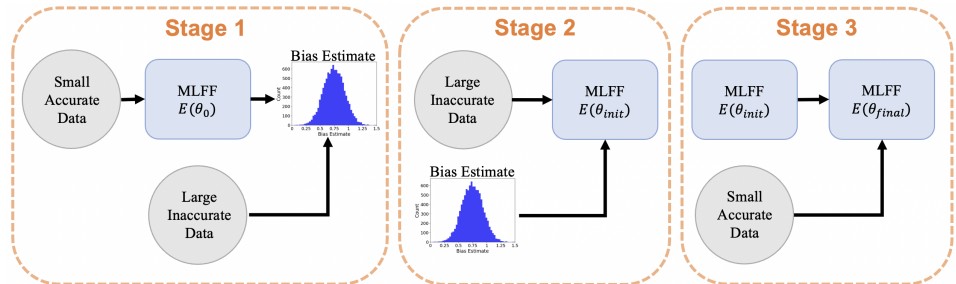

Figure 2: Asteroid workflow diagram.

To reduce the data generation cost needed to train MLFFs, we propose a multi-stage training framework, ASTEROID, to learn from a combination of both cheaply available inaccurate data and more expensive accurate data.

**Preliminaries.** For a molecule with $k$ atoms, we denote a configuration (the positions of its atoms in 3D) of this molecule as $x \in \mathbb{R}^{3k}$, its respective energy as $E(x) \in \mathbb{R}$, and its force as $F(x) \in \mathbb{R}^{3k}$. We denote the accurately labeled data as $\mathcal{D}_A = \{(x_1^a, e_1^a, f_1^a), ..., (x_N^a, e_N^a, f_N^a)\}$ and the inaccurately labeled data as $\mathcal{D}_I = \{(x_1^n, e_1^n, f_1^n), ..., (x_M^n, e_M^n, f_M^n)\}$, where $(x_i^a, e_i^a, f_i^a)$ represents the position, potential energy, and force of the $i$th accurately labeled molecule, (similarly $(x_j^n, e_j^n, f_j^n)$ for the $j$th inaccurately labeled data). Conventional methods train a force field model $E(\cdot; \theta)$ with parameters $\theta$ on the accurate data by minimizing the loss

$$\min_\theta \mathcal{L}(\mathcal{D}_A, \theta) = \frac{(1-\rho)}{3N} \sum_{i=1}^N \ell_f(f_i^a, \nabla_x E(x_i^a; \theta)) + \frac{\rho}{N} \sum_{i=1}^N \ell_e(e_i^a, E(x_i^a; \theta)), \qquad (1)$$

where $\ell_f$ is the loss function for the force prediction, and $\ell_e$ is the loss function for the energy prediction. Here the force is denoted by $\nabla_x E(x; \theta)$, i.e., the gradient of the energy $E(x; \theta)$ w.r.t. to the input $x$. In practice, most of the emphasis is placed on the force prediction, e.g. $\rho = 0.001$.

## 3.1 Bias Identification

The goal of ASTEROID is to leverage cheap MD simulation data to boost MLFF accuracy. However, the approximation algorithms used to generate cheap data $\mathcal{D}_I$ introduce a large amount of bias into some force labels $f^n$, which may significantly hurt accuracy. Motivated by this phenomenon, we aim to identify the most biased force labels so that we can avoid overfitting the bias during training. To do so, we use small amounts of accurately labeled data $\mathcal{D}_A$ to identify the levels of bias in the inaccurate dataset $\mathcal{D}_I$. Specifically, we train a force field model by minimizing $\mathcal{L}(\mathcal{D}_A, \theta)$ (Eq. 1), the loss over the accurate data, to get parameters $\theta_0$. Although the resulting model $E(\cdot; \theta_0)$ will not necessarily have good prediction performance because of the limited amount of training data, it can still help estimate the bias of the inaccurate data. For every configuration $x_j^n$ in the inaccurate dataset $\mathcal{D}_I$, we suspect it to have a large bias if there is a large discrepancy between its force label $f_j^n$ and the force label predicted by the accurately trained model $\nabla_x E(x_j^n; \theta_0)$. We can therefore use this discrepancy as a surrogate for bias, i.e. $B(x_j^n) = \|\nabla_x E(x_j^n; \theta_0) - f_j^n\|_1$.

## 3.2 Bias-Aware Training with Inaccurate Data

In the second stage of our framework, we train a force field model $E(\cdot; \theta_{\text{init}})$ from scratch on large amounts of *inaccurately labeled data* $\mathcal{D}_I$. Although this data can effectively capture the intrinsic

problem structure, the high levels of bias on some data points may propagate to the final model and harm generalization performance. To avoid over-fitting to the biased force labels, we use a bias-aware loss function that weighs the inaccurate data according to their bias. In particular, we use the weights $w_j = \exp(-B(x_j^n)/\gamma)$ for configuration $x_j^n$, where $\gamma$ is a hyperparameter to be tuned. In this way, low-bias points are given higher importance and high-bias points are treated more carefully. We then minimize the bias-aware loss function

$$\min_\theta \mathcal{L}_w(\mathcal{D}_I, \theta) = (1-\rho) \sum_{i=1}^{M} w_i \cdot \ell_f(f_i^n, \nabla_x E(x_i^n; \theta)) + \rho \sum_{i=1}^{M} w_i \cdot \ell_e(e_i^n, E(x_i^n; \theta)) \qquad (2)$$

to get parameters $\theta_{\text{init}}$, resulting in the initial estimate of the MLFF $E(\cdot; \theta_{\text{init}})$.

### 3.3 Fine-Tuning over Accurate Data

The model $E(\cdot; \theta_{\text{init}})$ contains information useful to the force prediction problem, but may still contain bias because it is trained on inaccurately labeled data $\mathcal{D}_I$. Therefore, we further refine it using accurately labeled data $\mathcal{D}_A$. Specifically, we use $E(\cdot; \theta_{\text{init}})$ as initialization for our final stage, in which we fine-tune the model over the accurate data by minimizing $\mathcal{L}(\mathcal{D}_A, \theta_{\text{final}})$ (Eq. 1). The full ASTEROID framework is illustrated in Figure 2.

## 4 ASTEROID for Unlabeled Data

In several settings, molecular configurations are generated without force labels, either because they are not generated via MD simulation (e.g. normal mode sampling, Smith et al. [2017]) or because the forces are not stored during the simulation [Köhler et al., 2022]. Although these unlabeled configurations may be cheaply available, they are not generated for the purpose of learning force fields and have not been used in existing literature. Here, we show that the unlabeled configurations can be used to obtain an initial estimate of the force field, which can then be further fine-tuned on accurate data. More specifically, we consider a molecular system where the number of particles, volume, and temperature are constant (NVT ensemble). Let $x$ refer to a molecule's configuration and $E(x)$ refer to the corresponding potential energy. It is known that $x$ follows a Boltzmann distribution, i.e.

$$p(x) = \frac{1}{Z} \exp\left(-\frac{1}{k_\beta T} E(x)\right),$$

where $Z$ is a normalizing constant, $T$ is the temperature, and $k_\beta$ is the Boltzmann constant. In practice, configurations generated using normal mode sampling [Unke et al., 2021] or via a sufficiently long NVT MD simulation follow a Boltzmann distribution.

Recall that we model the energy $E(x)$ as $E(x; \theta)$, and the force can be calculated as $F(x; \theta) = \nabla_x E(x; \theta)$. It follows from Hyvärinen [2005] that we can learn the score function of the Boltzmann distribution using score matching, where the score function is defined as the gradient of the log density function $\nabla_x \log p(x)$. In our case, we observe that the force on a configuration $x$ is proportional to the score function, i.e., $F(x) \propto \nabla_x \log p(x)$. Therefore, we can use score matching to learn the forces by minimizing the unsupervised loss

$$L(\theta) = \mathbb{E}_{p(x)} \left[ \frac{1}{\beta} \text{Tr}[\nabla_x F(x; \theta)] + \frac{1}{2} ||F(x; \theta)||^2 \right], \qquad (3)$$

where $\beta = \frac{1}{k_\beta T}$. A derivation can be found in Appendix A.5. Although this objective allows us to solve the force matching problem in an unsupervised manner, the unsupervised loss is difficult to optimize in practice. To reduce the cost of solving Eq. 3, we adopt sliced score matching [Song et al., 2020]. Sliced score matching takes advantage of random projections to significantly reduce the cost of solving Eq. 3, allowing us to apply score matching to large neural models such as GemNet.

In our experiments, we find that score matching does not match the accuracy of CCSD(T) force labels. Instead, we can think of score-matching as a form of inaccurate training. We therefore use score matching as an alternative to stages one and two of the ASTEROID framework. That is, we minimize Eq. 3 to get $\theta_{\text{init}}$, after which the model is fine-tuned on the accurate data.

# 5    Experiments

For our main experiments, we evaluate ASTEROID on MLFF datasets and downstream MD simulation tasks. For ASTEROID, we consider three settings: using DFT data to enhance CCSD(T) training, using empirical force field data to enhance CCSD(T) training, and using unlabeled configurations to enhance CCSD(T) training. In each setting, we evaluate the performance of ASTEROID and standard training over a variety of data generation budgets.

## 5.1    Datasets and Models

For the CCSD(T) data, we use MD17@CCSD, which contains 1,000 configurations labeled at the CCSD(T) and CCSD level of accuracy for five molecules [Chmiela et al., 2017]. For DFT data, we use the MD17 dataset, which contains molecular configurations labeled at the DFT level of accuracy [Chmiela et al., 2017]. For the empirical force field data, we generate 100,000 configurations for each molecule using the OpenMM empirical force field software [Eastman et al., 2017]. For the unlabeled datasets, we use MD17 with the force labels removed.

The MD17 datasets do not release the computational cost of data generation, but when we replicate their experiments, we find that CCSD(T) labels cost roughly 40 times more than DFT labels. However, the difference in cost between CCSD(T) and DFT labels may change drastically depending on the implementation of each method. Therefore we evaluate the performance of ASTEROID when CCSD(T) force labels are 20, 40, and 80 times more expensive than DFT force labels. Note that the cost of empirical force labels is essentially negligible (more than 10,000 times cheaper) compared to CCSD(T) labels [Folmsbee and Hutchison, 2021].

In each setting, we compare standard training with 250, 450, 650, or 850 CCSD(T) training samples with ASTEROID. For ASTEROID, we use either 1000, 2000, or 4000 DFT datapoints (corresponding to cost ratios of 20:1, 40:1, and 80:1 for DFT and CCSD(T) labels), and 200, 400, 600, or 800 CCSD(T) data points. The computational budget of standard training and ASTEROID are therefore equivalent. A validation set of size 50 and a test set of size 500 are used in all experiments.

We implement our method on GemNet and EGNN. For GemNet we use the same model parameters as Gasteiger et al. [2021]. For EGNN, we use a 5-layer model and an embedding size of 128. When training with inaccurate data, we train with a batch size of 16 and stop training when the loss stabilizes. In the fine-tuning stage, we use a batch size of 10 and train for a maximum of 2000 epochs. To tune the bias aware loss parameter $\gamma$, we search in the set $\{0.1, 0.5, 1.0, 2.0\}$ and select the model with the lowest validation loss. Comprehensive experimental details are deferred to Appendix A.6.

## 5.2    Enhancing Force Fields with DFT

We display the results for using DFT data to enhance CCSD(T) training in Figure 3 for GemNet and Figure 4 for EGNN. From these figures, we can see that ASTEROID can outperform standard training for all amounts of data and cost ratios. Using larger amounts of inaccurate data can significantly reduce prediction error, but the 20:1 cost ratio already has large performance gains over standard training. When applied to GemNet in low resource settings, ASTEROID reduces the average prediction error by 39.4% and improves sample efficiency by a factor of 2. For EGNN, ASTEROID improves prediction error by 56% and increases sample efficiency by more than 3 times. The large performance increase for EGNN may be due to the fact that the EGNN architecture has less inductive bias than GemNet, and therefore may struggle to learn the structures of the underlying force field with only a small amount of data.

## 5.3    Enhancing Force Fields with Empirical Force Calculation

We present the results for empirical force field in Table 1. Additional results for GemNet can be found in Appendix A.7. Again we find that ASTEROID significantly outperforms the supervised baseline, improving prediction accuracy by 36% for GemNet and by 17% for EGNN. The good performance on empirical force fields indicates that ASTEROID is relatively robust to the label noise on the inaccurate data.

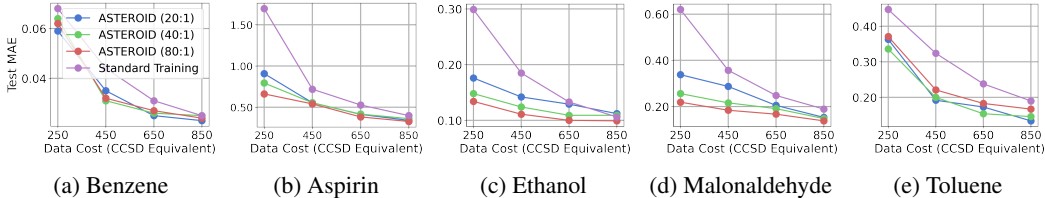

| (a) Benzene | (b) Aspirin | (c) Ethanol | (d) Malonaldehyde | (e) Toluene |

Figure 3: Main results for GemNet when DFT data is viewed as inaccurate. The ratio refers to the number of DFT calculations that are equivalent to one CCSD(T) calculation. The results are measure in kcal/mol/Å, averaged across dimensions and atoms.

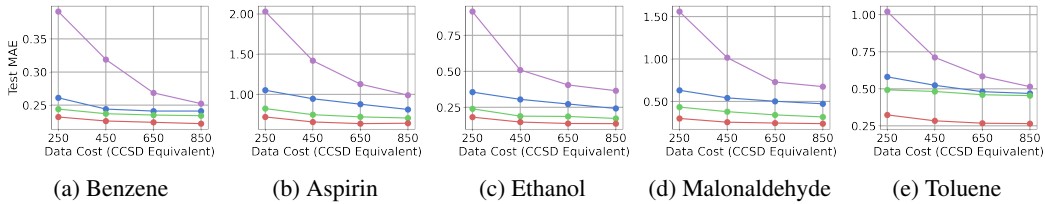

| (a) Benzene | (b) Aspirin | (c) Ethanol | (d) Malonaldehyde | (e) Toluene |

Figure 4: Main results for EGNN when DFT data is viewed as inaccurate.

Table 1: Test MAE of ASTEROID with empirical force field data. The results are measure in kcal/mol/Å, averaged across dimensions and atoms. The training set for the fine-tuning stage contains 200 molecules labeled at the CCSD(T) level. "Malo." refers to malonaldehyde and "Standard Tr." refers to standard training.

|  | **Aspirin** | **Benzene** | **Malonaldehyde** | **Toluene** | **Ethanol** |
|---|---|---|---|---|---|
| **GemNet** | | | | | |
| Standard Training | 1.554 | 0.083 | 0.801 | 0.591 | 0.348 |
| ASTEROID | **0.843** | **0.048** | **0.516** | **0.337** | **0.301** |
| **EGNN** | | | | | |
| Standard Training | 1.897 | 0.297 | 1.466 | 0.777 | 0.840 |
| ASTEROID | **1.314** | **0.268** | **1.341** | **0.664** | **0.637** |

## 5.4 Enhancing Force Fields with unlabeled Molecules

We first verify that our proposed score matching approach can learn the forces on unlabeled molecules by comparing the prediction accuracy of models trained by score matching with models trained on supervised data (DFT and empirical force fields). We measure prediction accuracy on CCSD(T) datasets and show the results in Figure 7. Surprisingly, we find that the prediction error of score matching is between that of DFT and empirical force fields. This indicates that relatively accurate force predictions can be obtained by only solving the unsupervised loss in Eq. 3.

Next we apply ASTEROID to settings where unlabeled data is available by fine-tuning the model obtained from score matching. We present the results in Table 2, where we find that ASTEROID can improve prediction accuracy by 18% for GemNet and 4% for EGNN. If unsupervised data can be generated cheaply (i.e. through normal mode sampling), then our approach can be used to boost the performance of MLFFs with little additional cost.

## 6 Discussion

**Related Work.** There are several works which we compare ASTEROID with.

⋄ **Δ-ML** [Ramakrishnan et al., 2015, Bogojeski et al., 2020], learns the difference between inaccurate (DFT) and accurate (CCSD(T)) force predictions, therefore speeding up MD simulation while

Table 2: Accuracy of ASTEROID with unlabeled molecular configurations. The results are measure in kcal/mol/Å, averaged across dimensions and atoms. The training set for the fine-tuning stage contains 200 CCSD(T) labeled molecules.

|  | Aspirin | Benzene | Malonaldehyde | Toluene | Ethanol |
|---|---|---|---|---|---|
| **GemNet** | | | | | |
| Standard Training | 1.554 | **0.083** | 0.801 | 0.591 | 0.348 |
| ASTEROID | **0.928** | 0.093 | **0.629** | **0.475** | **0.314** |
| **EGNN** | | | | | |
| Standard Training | 1.897 | **0.297** | 1.466 | 0.777 | 0.840 |
| ASTEROID | **1.756** | 0.305 | **1.382** | **0.740** | **0.823** |

maintaining high accuracy. However, this approach requires a DFT calculation to be done during inference, greatly increasing inference time compared to ASTEROID or standard MLFFs [Folmsbee and Hutchison, 2021].

⋄ **ANI-1ccx** [Smith et al., 2019, Deringer et al., 2020] train an MLFF on a huge DFT dataset comprised of many molecules, and then finetune on many CCSD(T) labeled molecules with a goal of learning a general MLFF. Notably, the method from Smith et al. [2017] only trains on equilibrium states and may not work well for MD trajectory data. To compare ANI-1ccx with ASTEROID, we evaluate the provided model checkpoint in the zero-shot setting (as in [Smith et al., 2019]) and when finetuned on each MD17 molecule. Note that the data generation cost of ANI-1ccx is much more expensive than ASTEROID, using 2,500 times more CCSD(T) data and 500 times more DFT data.

⋄ **sGDML** [Chmiela et al., 2019] is a kernel-based MLFF method that can perform well when limited training data is available by incorporating relevant physical constraints into the MLFF.

As can be seen in Table 3, ASTEROID trained MLFFs can achieve lower test errors than all of the baselines except $\Delta$-ML. However, since $\Delta$-ML requires a DFT calculation during inference, MD simulation with $\Delta$-ML will take 100 to 1000 times longer than with ASTEROID [Folmsbee and Hutchison, 2021, Gasteiger et al., 2021]. Therefore ASTEROID results in the most useful force fields out of all the baselines, while having a smaller or equivalent data generation cost.

Table 3: Accuracy of ASTEROID compared with competitive baselines with a data budget of 250 CCSD(T) points. FT refers to fine-tuning ANI-1ccx on MD17@CCSD. The model is GemNet.

|  | Aspirin | Benzene | Malonaldehyde | Toluene | Ethanol |
|---|---|---|---|---|---|
| **ANI-1** | 1.897 | 0.297 | 1.466 | 0.777 | 0.840 |
| **ANI-1 (FT)** | 1.314 | 0.268 | 1.341 | 0.664 | 0.637 |
| **$\Delta$-ML** | **0.801** | — | **0.182** | 0.350 | — |
| **sGDML** | 1.727 | 0.097 | 0.923 | 0.478 | 0.902 |
| **ASTEROID** | 0.908 | **0.059** | 0.338 | **0.306** | **0.176** |

**Ablation.** We conduct a detailed ablation study in Appendix A.3, which shows ASTEROID is fairly robust to hyperparameter selection.

**MD Simulation** We show the results of MD simulation results in Appendix A.2, where we observe ASTEROID can result in stable simulation.

**Asteroid with Multiple Molecules.** We try ASTEROID on multiple molecules simultaneously in Appendix A.1. We find mixed results, indicating this could be an exciting direction to explore further.

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

# A Appendix

## A.1 ASTEROID with Multiple Molecules

Table 4: Accuracy of ASTEROID when the inaccurate data is comprised of multiple molecules.

|                    | Aspirin | Benzene | Malonaldehyde | Toluene | Ethanol |
|--------------------|---------|---------|---------------|---------|---------|
| Standard Training  | 1.554   | 0.074   | 0.776         | 0.566   | 0.351   |
| ASTEROID (Multi)   | **0.716** | **0.05** | 0.480       | **0.237** | 0.269 |
| ASTEROID           | 0.908   | 0.059   | **0.338**     | 0.306   | **0.176** |

To explore the generality of ASTEROID, we further investigate the setting where the inaccurate data for ASTEROID is comprised of multiple molecules. After training such a general purpose (but inaccurate) MLFF, we separately fine-tune the MLFF on each of the MD17 molecules labeled at the CCSD(T) level of accuracy. This setting is very intriguing, since it means that only one network must be pre-trained per molecule. This approach could potentially allow for a large reduction in the memory requirement and pre-training time of ASTEROID.

The results for a total budget of 250 CCSD(T) data points can be seen in Table 4. From Table 4 we can see that training ASTEROID over multiple molecules can significantly reduce the test error compared to standard training. On Aspirin, Benzene, and Malonaldehyde, ASTEROID trained over multiple molecules can perform better than ASTEROID for just a single molecule, likely due to the fact that these molecules all share common structures. However for Malonaldehyde and Ethanol, training over multiple molecules harms performance. Given the mixed performance and the simplicity of single molecule pre-training, it is expected that single molecule pre-training would be favored in most scenarios.

## A.2 MD simulation

It has been observed that low test errors are not sufficient for obtaining stable MD simulation dynamics [Stocker et al., 2022]. To ensure that ASTEROID can be used for MD simulations, we evaluate the performance of MLFFs trained by ASTEROID in downstream MD simulation tasks. First, we demonstrate that ASTEROID-trained MLFFs can produce stable dynamics, while MLFFs trained on DFT data and empirical force fields diverge. Using the Atomic Simulation Environment (ASE) [Larsen et al., 2017], we simulate the behavior of a benzene molecule using forces calculated by a MLFF trained with ASTEROID, an MLFF trained on DFT data only, and the Lennard-Jones empirical force field. We simulate the molecule with Langevin dynamics, where the steps size is 0.5 femtoseconds, the temperature is 500K, the friction coefficient is 0.002, and the maximum number of time steps is 10000. The results of these simulations can be seen in Figure 5a, where ASTEROID is able to produce stable dynamics. On the other hand, the error compounding of the DFT trained MLFF and the Lennard-Jones potential results in diverged simulations and unlikely molecular configurations.

We also compare the MD simulations generated using ASTEROID with those generated using standard training, where both MLFFs are trained with a data budget equivalent to 250 CCSD(T) points. Inspired by Stocker et al. [2022], we run MD simulations with varying step sizes on the aspirin molecule to evaluate robustness. In Figure 5b we plot the proportion of simulations that converge with varying simulation step sizes. We define a simulation as converged if the maximum pairwise distance between atoms remains within a specified threshold. For each step size, we report the result over 20 Langevin dynamics simulations, each with a length of one picosecond. The ASTEROID framework is able to maintain steady performance across step sizes, and almost all the simulations converge. In contrast, the simulations powered by standard MLFFs fail with larger step sizes.

Figures 5a and 5b show the advantages of ASTEROID go beyond reducing test error and allow for stable simulations to be run over 3 times as fast as standard MLFFs. Interestingly, Stocker et al. [2022] find that to train robust MLFFs, much more training data than the amount needed for low test

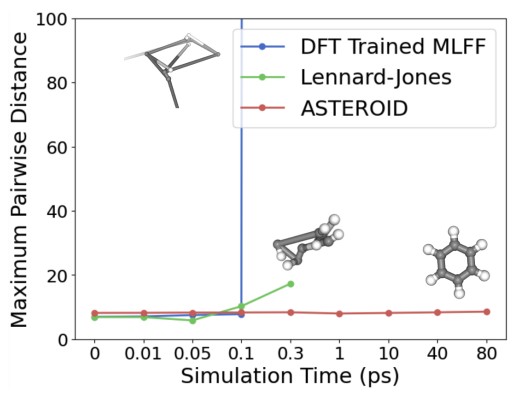

(a) Benzene molecule during MD simulation.

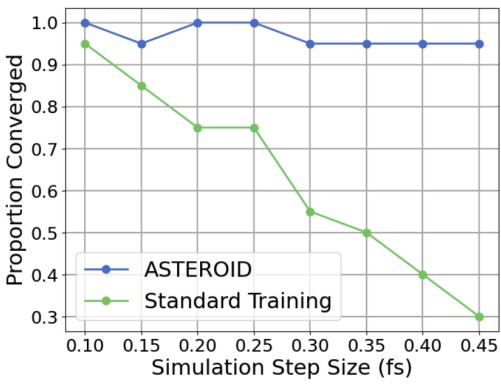

(b) Proportion of converged simulations (Aspirin).

Figure 5: MD simulation analysis for ASTEROID.

error should be used. ASTEROID provides a cost-efficient way to increase the size of the dataset, therefore enhancing robustness at a low data cost.

## A.3 Analysis

◇ **Ablation Study** We study the effectiveness of each component of ASTEROID. Specifically, we investigate the importance of bias-aware training (BAT) and fine-tuning (FT) when compared with standard training. The results for Gemnet can be seen in Table 5. As shown in Table 5, each of ASTEROID's components is effective and complementary to one another. We find that bias-aware training is most helpful with GemNet, where it reduces test error by 6.5% on average, possibly due to the fact that GemNet has more capacity to overfit harmful data points than EGNN.

Table 5: Ablation study for ASTEROID on Gemnet. The inaccurate data is DFT labeled configurations and the accurate dataset contains 200 CCSD(T) labeled configurations. "AST." refers to ASTEROID.

|  | Aspirin | Benzene | Malonaldehyde | Toluene | Ethanol |
|---|---|---|---|---|---|
| Standard Training | 1.554 | 0.074 | 0.776 | 0.566 | 0.351 |
| AST. w/o FT | 4.670 | 3.252 | 2.726 | 3.342 | 5.107 |
| AST. w/o BAT | 1.095 | 0.064 | 0.347 | 0.309 | 0.183 |
| ASTEROID | **0.908** | **0.059** | **0.338** | **0.306** | **0.176** |

◇ **Sensitivity** We also investigate the sensitivity of ASTEROID to the hyperparameters $\gamma$. We use a data budget of 250 CCSD(T) points. From Figure 6a we can see that ASTEROID is robust to the choice of hyperparameters, outperforming standard training in every setting.

◇ **Size of inaccurate data.** To demonstrate that ASTEROID can exploit varying amounts of inaccurate data, we plot the performance of ASTEROID with different cost ratios. This can be seen in Figure 6b, where a budget of 250 CCSD(T) points is used. ASTEROID performs best when large amounts of inaccurate data are available but still increases the accuracy by 20% when the cost ratio is small.

## A.4 Accuracy of Score Matching

## A.5 Derivation of Score Matching for Forces

For a given molecule with conformations $x_1, .., x_n$, let us denote energy as $E(x)$. Then the the Boltzmann/Equilibrium distribution for the molecule is given by

$$p(x) = \frac{1}{Z}\exp(-\beta E(x)),$$

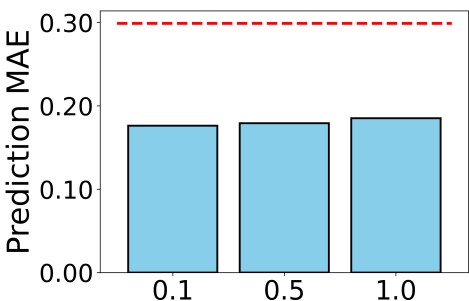
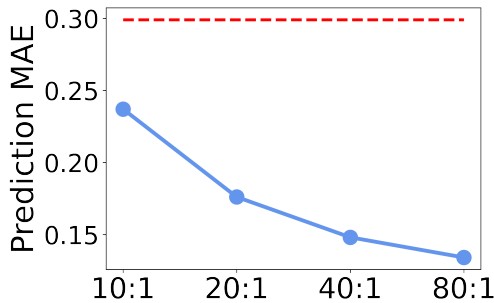

(a) Sensitivity study for $\gamma$ (ethanol). The red line represents standard training.

(b) Performance with different cost ratios between DFT and CCSD(T) (ethanol).

Figure 6: Ablation and sensitivity studies for ASTEROID.

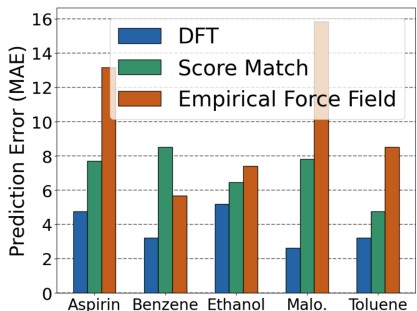

Figure 7: Prediction errors of models tested on CCSD(T) data. Models are not fine-tuned on the CCSD(T) data.

where $Z$ is a normalizing constant, $\beta = \frac{1}{k_\beta T}$, $k_\beta$ is the Boltzmann constant, and $T$ is the temperature under which the simulation is run. Then we can see that the force on a conformation $x$ is equivalent to the score, i.e. $F(x) = -\nabla_x E(x) = \frac{1}{\beta}\nabla_x \log p(x)$. Therefore learning the force $F(x)$ is equivalent to learning the score $\frac{1}{\beta}\nabla_x \log p(x)$. Suppose we parameterize the MLFF to directly predict the force as $F_\theta(x)$. Then the force matching loss can be written as

$$\mathcal{L}(\theta) = \frac{1}{2}E_{x\sim p(x)}\|F_\theta(x)-F(x)\|_2^2 = \frac{1}{2}E_{x\sim p(x)}\|F(x)\|_2^2 - E_{x\sim p(x)}\left[\langle F_\theta(x), F(x)\rangle\right] + \frac{1}{2}E_{x\sim p(x)}\|F_\theta(x)\|_2^2.$$

The middle term can then be expanded as

$$E_{x \sim p(x)} \left[ \langle F_\theta(x), F(x) \rangle \right] = \int_x p(x) \langle F_\theta(x), F(x) \rangle dx \qquad \text{Integration over x.}$$

$$= \int_x p(x) \sum_{i=1}^d (\frac{1}{\beta} \frac{d\log p(x)}{dx_i} F_\theta(x)_i) dx \qquad \text{Expansion of inner product.}$$

$$= \frac{1}{\beta} \sum_{i=1}^d \int_x \frac{dp(x)}{dx_i} F_\theta(x)_i dx \qquad \text{Simplify and move summation.}$$

$$= \frac{1}{\beta} \sum_{i=1}^d \int_{x_{i-}} \int_{x_i} F_\theta(x)_i dp(x) d_{x_{i-}} \qquad \text{Integrate over } x_i.$$

$$= \frac{1}{\beta} \sum_{i=1}^d \int_{x_{i-}} \left( F_\theta(x)_i dp(x)|_{-\infty}^{+\infty} - \int_{x_i} p(x) dF_\theta(x)_i \right) d_{x_{i-}} \quad \text{Partial inegration.}$$

$$= -\frac{1}{\beta} \sum_{i=1}^d \int_{x_{i-}} \int_{x_i} p(x) \frac{dF_\theta(x)_i}{dx_i} dx_i d_{x_{i-}} \quad \text{Normality assumption.}$$

$$= -\frac{1}{\beta} \sum_{i=1}^d E_{x \sim p(x)} \left[ \frac{dF_\theta(x)_i}{dx_i} \right] = -\frac{1}{\beta} E_{x \sim p(x)} \left[ \text{Tr} \left[ \nabla_x F_\theta(x) \right] \right].$$

Therefore we have the loss

$$\mathcal{L}(\theta) = E_{x \sim p(x)} \left[ \frac{1}{\beta} \text{Tr} \left[ \nabla_x F_\theta(x) \right] + \frac{1}{2} \| F_\theta(x) \|_2^2 \right].$$

The first term in the loss disappears as it is not dependent on $\theta$.

## A.6  Experimental Details

In this section, we go over the experimental details.

**GemNet Training Details.** To train the bias identification method, we train a freshly initialized model with a batch size of 10 on the accurate dataset for 2000 epochs. To train the inaccurate model, we train a freshly initialized model with the bias aware loss function and batch size 16 over the inaccurate dataset. Finally, to finetune the inaccurately trained model, we train a model with a batch size of 10 on the accurate dataset for 2000 epochs. In each stage of training, we use the following hyperparamers:

- Evaluation Interval: 1 epoch
- Decay steps: 1200000
- Warmup steps: 10000
- Decay patience: 50000
- Decay cooldown: 50000

The rest of the parameters are the same as used in Gasteiger et al. [2021].

**EGNN Training Details.** The EGNN training setup is similar to GemNet. To train the bias identification method, we train a freshly initialized model with a batch size of 10 on the accurate dataset for 2000 epochs. To train the inaccurate model, we train a freshly initialized model with the bias aware loss function and batch size 32 over the inaccurate dataset. Finally to finetune the inaccurately trained model, we train a with a batch size of 10 on the accurate dataset for 2000 epochs. In each stage of training we use the following hyperparamers:

- Evaluation Interval: 1 epoch

   • Learning rate: $10^{-4}$ for inaccurate training, $10^{-5}$ for finetuning

451   • num_layers: 5

452   • embedding_size: 128

## A.7   Additional Results

453

454   Here we include additional results for ASTEROID when empirical force field data is viewed as
455   inaccurate. For the baseline model we use GemNet. The ASTEROID framework again leads to
456   consistent gains across all amounts of data.

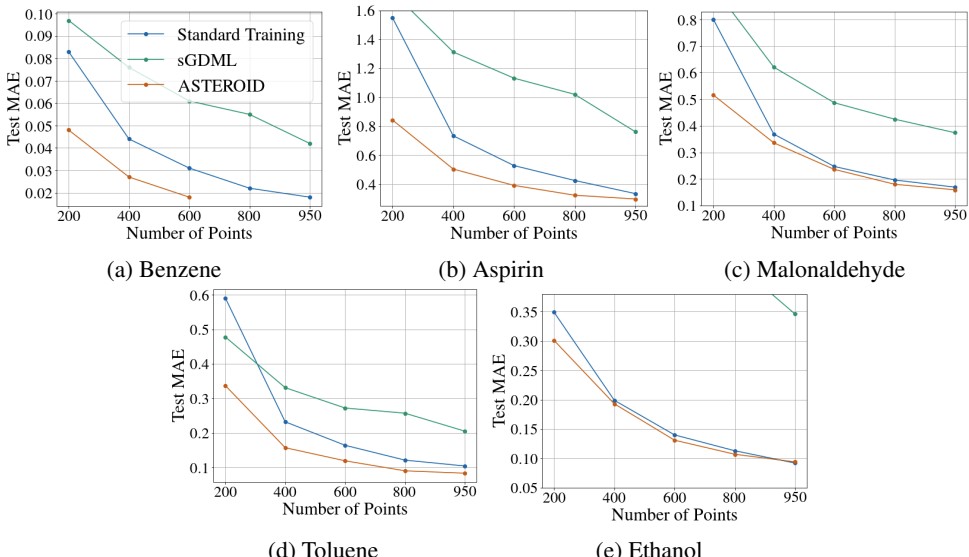

Figure 8: Main results for GemNet when empirical force field data is viewed as inaccurate.

## A.8   Baseline Implementations

457

458   ◇ **ANI1-ccx.** In order to have a fair comparison with Smith et al. [2019], we consider two ANI-1ccx
459   based baselines. In the first baseline, we take the provided ANI-1ccx checkpoint and analyze its zero-
460   shot performance on the MD17 dataset. For the second ANI-1ccx baseline, we finetune ANI-1ccx
461   separately on each molecule in MD17 until the validation loss has converged.

462   ◇ **$\Delta$-ML for GemNet.** For a fair comparison with ASTEROID, we implement $\Delta$-ML on GemNet
463   and the MD17 molecules. Given a molecular configuration $x$, it's corresponding DFT force labels $f^i$,
464   and the CCSD(T) force labels $f^a$, optimize the supervised loss

$$\min_\theta \mathcal{L}_\Delta(x, \theta) = \ell_f(f^a, f^i + \nabla_x E(x; \theta)). \tag{4}$$

465   We then optimize this loss over all train configurations for a given molecule, using an energy loss
466   similar to (1). During inference, we predict the CCSD(T) force labels as $f^i + \nabla_x E(x; \theta)$, which
467   requires the DFT force label to be computed.

468   Since the mapping between DFT labeled configurations and MD-17 labeled configurations is not
469   explicitly given, we must find it ourselves. For every point in the CCSD(T) dataset, we find the
470   closest point to it in the DFT dataset. For each of the molecules listed in Section 6, the difference
471   between the CCSD(T) configuration and closest DFT configuration is $1 \times 10^{-5}$. For Benzene and
472   Ethanol, we find that such a mapping is not available.

### A.9  ASTEROID Toy Example

We have added a new result using a two-layer MLP with 128 hidden units each and synthetic data. This experiment shows that ASTEROID can significantly improve generalization error in more general settings. In this experiment, we generate a biased dataset of 2000 points according to $Y = AX + b$, where where $X \sim N(0, 1)$ has dimension 16, $b$ is the bias, and $A$ is a randomly generated Gaussian matrix of dimension $16 \times 16$. The bias b is chosen uniformly from the set $[0, 2, 4, 8, 16]$. We also generate varying levels of accurate data according to $Y = AX$, where $X \sim N(0, 1)$. We then evaluate the test MAE of ASTEROID and standard training over a variety of accurate data sizes. We find that ASTEROID significantly outperforms standard training.

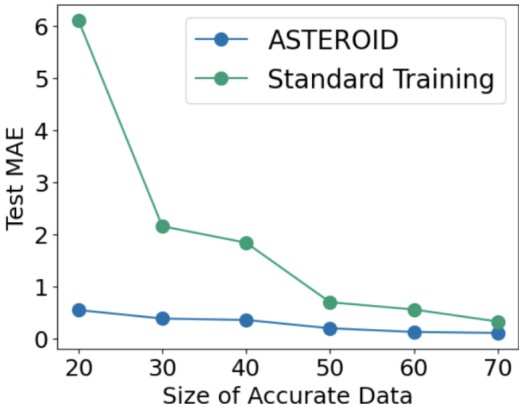

Figure 9: Asteroid toy example.