# OpenReview forum: "Machine Learning Force Fields with Data Cost Aware Training"
_NeurIPS.cc/2023/Workshop/AI4Science — NeurIPS2023-AI4Science Poster_

### Official Review · Reviewer_DGuw · 2023-10-22
**Machine Learning Force Fields with Data Cost Aware Training**

**Rating:** 9
**Confidence:** 3

**Review:**

The authors propose a multi-fidelity approach for training MLFFs for molecular simulation. The model is first trained on low-accuracy DFT data (biased by a surrogate model trained on high accuracy data), then fine tuned with high accuracy CCSD(T) data. The results show improved performance with the approach compared to just CCSD(T) data in the standard training. However, the magnitude of improvement seems to vary a lot by chemical system. Nonetheless, the work presents a useful approach when limited in accurate data.

They also propose a unsupervised learning approach based on a scoring function from the boltzmann distribution, which ultimately resulted in accuracies less than DFT. This is another useful approach if computational expenses are very limited.

The questions are well posed and experiments well designed.

---

### Meta-Review · Area_Chair_nDvU · 2023-10-27

**Recommendation:** Accept (Oral)
**Confidence:** 3

**Metareview:**

The paper presents a multi-fidelity approach for training MLFFs in molecular simulation, using high and low-accuracy data to reduce computational costs. The results demonstrate improved performance, and the method proves valuable when accurate data is insufficient.
The research questions and experiment designs are well-structured, and the only concern raised is the variability in the magnitude of improvement across different chemical systems.